# Model-Based Adaptive Collaboration of Multi-Terminal Internal Force Tracking

**Zhala Wang** [1,2], **Jingmin Dai** [1,*] **and Fei Song** [3]

1  School of Instrumentation Science and Engineering, Harbin Institute of Technology, Harbin 150001, China; wzl@oit.edu.cn
2  Department of Information Engineering, Ordos Institute of Technology, Ordos 017000, China
3  College of Art and Design, Wuhan Institute of Technology, Wuhan 430223, China; sf050402@163.com
*  Correspondence: djm86415146@163.com; Tel.: +86-157-5069-7363

**Abstract:** This paper proposes a multi-terminal adaptive collaborative operation method for solving the problem of unstable internal force tracking in the clamping and handling of unknown objects by multi-terminal robots. In the proposed method, the internal command force changes the complex internal force control problem into an internal force tracking problem from multi-slave to master. Moreover, we develop an algorithm for multi-slave setups to estimate the object stiffness and motion uncertainty in the direction of the internal command force according to Lyapunov theory. Finally, the impedance control generates a reference trajectory for the multi-slave to maintain the desired internal force and track the master's motion. Several experiments were conducted on a self-made robot. The experimental results show that the oscillation amplitude of each slave end is less than 1 mm and the directional oscillation amplitude is less than 1 degree during the tracking of the desired commanded internal force. For objects with a low stiffness, the error of the commanded internal force is less than 1 N (6%) per slave. The error in tracking the commanded internal force for objects with a high stiffness is less than 2 N (8%). The results prove the feasibility and effectiveness of the proposed method.

**Keywords:** multi-terminal; force tracking; adaptive

## 1. Introduction

Collaborative force closed collaboration can perform more complex and uncertain tasks in a wide range of application scenarios, such as rapid and stable handling or large object rescue in nonstructural hazardous environments [1–4]. These cooperative systems form a closed operation chain and form a group of motion constraints through contact forces. Thus, the degree of freedom of the closed operation chain system is significantly reduced to form a task chain. During this process, the complex time-varying internal forces generated by the dynamic system formed by various force and motion transmission components (referred to as terminals in this article) and objects are crucial for the stability of the task. Effective control of the workspace is crucial. Due to the existence of motion uncertainty and force uncertainty, the controller design of closed cooperative systems with multiple terminals cooperating in force becomes more complex and unstable. At present, in order to solve the kinematics uncertainty when manipulating unknown objects in the range of dual terminals, researchers have proposed several solutions using advanced control theory.

One example of this is the use of sliding mode control to improve the trajectory-tracking accuracy [5,6]. An adaptive control scheme [7,8] was proposed for the cooperative control of multi-arm robots in the case of unknown object information. The concept of absolute relative motion was proposed for the application scenario of grasping (force closure within the fixture) to move large objects [9,10].

A time-varying internal force control scheme includes impedance control and hybrid position/force control. However, whether this is executed by adaptive control, proposed in [11,12] to improve the time-varying force-tracking performance of objects with unknown stiffness, or by hybrid position/force control, proposed in [13–16] with the decoupling task space as the position and force subspaces, to a certain extent, the scheme depends on the known environment information and kinematics model. A method was proposed in [17] to solve the time optimal path tracking problem of two robots executing collaborative grasping tasks. The time optimal path was determined by dynamic programming, and a task space admittance control scheme was used to generate a contact model. This method is suitable for grasping general objects in surface contact with robots. However, the problem of time-varying internal force tracking for uncertain targets has not been resolved [18]. After analyzing the end effector and environment of the robot, the contact force was used as the feedback force of a position-based impedance controller to actively track the desired dynamic uncertain environmental force. In order to reduce the force-tracking error caused by environmental position uncertainty, variable impedance control based on online adjustment of impedance parameters was proposed to compensate for the unknown environment and dynamic expected force. The introduction of variable-stiffness adaptive impedance control to adjust internal forces, which was proposed in an adaptive hybrid impedance control method for dual-arm robots [19], is the most valuable reference related to this work. However, variable-stiffness adaptive impedance control for robot terminals requires high control parameter settings, which can easily lead to system instability.

These methods, which to some extent adapt to uncertain environments, improve the robustness of robot systems to uncertain dynamic environmental forces in collaborative force closed collaboration. However, such research is still in the exploration and improvement stage, and more experiments and methods are needed to improve its performance and applicability for numerous application scenarios. In both static- and weak-dynamic-force environments, the force-tracking performance of robots has been improved, enabling better tracking of target forces [20–22]. In the face of more complex models, complex dynamic forces, and unknown environments, the robustness of the algorithm still needs to be further strengthened to cope with uncertainty and unknowns.

This paper aims to solve the time-varying complex internal force-tracking and control problem for strong, stable clamping and handling of unknown objects through a multi-terminal system. Specifically, in the proposed solution, the latter system is based on multi-terminal internal force workspace transformation, an adaptive control algorithm for time-varying force tracking from multiples slaves to a single master. Moreover, we study the estimation of uncertain and unknown motion and the adaptive reference trajectory generation method using absolute relative motion strategies.

The proposed clamping and handling model-based multi-terminal internal force transformation and the adaptive collaboration scheme are simple and robust for operating unknown arbitrary objects. The overall control process simulates human gripping and handling of unknown objects. Compared to the existing literature, the main contributions of this paper can be summarized as follows:

1.  Based on the absolute relative motion strategy, we propose the time-varying force-tracking control mode and command internal force concept of a contact single-master–multi-slave terminal. Furthermore, the time-varying complex internal force working space is simplified and transformed according to the concept of command internal forces.
2.  A method for estimating the unknown stiffness and motion of the direction of multi-internal force commands is proposed. The suggested method is simple and robust to object stiffness, shape changes, and the uncertain motion of contact points.
3.  Based on the estimation of stiffness and motion in the command internal force direction, we use the uncertain motion and force of impedance model in the command internal force direction to track the formation method of the adaptive reference trajectory. This means that the force of the contact point measured by the force sensor and kinematics can stabilize the collaborative task without building a dynamic object model.

The remainder of this paper is organized as follows: Section 2 describes a complex time-varying internal force workspace comprising multi-terminals and absolute and relative motion control. Section 3 introduces the methods of estimating the unknown dynamic parameters in the command internal force direction and the adaptive reference trajectory formation. Section 4 evaluates the proposed algorithms on a self-made multi-terminal collaborative robot platform. Finally, Section 5 concludes this paper and discusses future research directions.

## 2. Model Establishment of a Multi-Terminal Clamping System

### 2.1. Formalizing the Workspace

The following symbols represent the position, velocity, and force of each system part used throughout the paper. Note that in the symbols below, the superscripts and subscripts are omitted.

$h$      Vector composed of $F$ and $N$.
$F$      Force vectors of various parts of the system.
$N$      Torque vector of each part of the system.
$q$      A vector composed of $h$ vectors from multiple terminals.
$O$      Origin of coordinate systems for various parts of the system.
$v$      Translation velocity vector of various parts of the system.
$p$      Position vectors of various parts of the system.
$Q$      Direction vectors of various parts of the system.
$l$      Virtual rod vector from the coordinate origin $O_{hi}$ of each terminal to the object coordinate origin $O_o$.
$\omega$      Angular velocity vectors of various parts of the system.

The following subscripts are defined to distinguish the position, velocity, and force vectors of each part of the system.

$o$      Position, velocity, and force related to objects.
$li$      Position, velocity, and force of the $i$th virtual rod tip.
$hi$      Position, velocity, and force at the root of the virtual rod (at the $i$th contact terminal).
$hj$      The position and speed of the root of the virtual rod (at the $i$th slave terminal) after system simplification.
$i$      Contact point or terminal serial number.
$j$      Represents the simplified serial numbers of each slave terminal, $j = a, b, c$.
$r$      Relative (internal) position, velocity, and force.
$m$      Generalized relative (internal) position, velocity, and force vectors.
$e$      Measure of position, velocity, and force.

The following superscript is defined in the upper left corner of the vector to distinguish the coordinate system represented by the vector.

$w$      Robot coordinate system $\Sigma_w$.
$hi$      Simplified coordinate systems for each terminal $\Sigma_{hi}, i = a, b, c, d$.
$hj$      Simplified coordinate systems for each slave terminal $\Sigma_{hj}, j = b, c, d$.

Please refer to the text for other symbols not defined here.

This paper considers the process of a humanoid remote-operation robot equipped with four end effectors to manipulate objects with unknown information other than visual information. Figure 1 illustrates each terminal, which has two customized optical bionic force sensors. These four terminals cooperate to stably clamp unknown objects and maintain adaptive stable handling without relative sliding during any collaboration of the remote operation or the planning path. This paper makes the following assumptions:

1. Remote operation enables each force-sensing contact point installed at the terminal to reach a reasonable, cooperative position on the object's surface, and all contact points contact the object's surface.

2.  The remote operation causes the four terminals to form an origin $O_o$ in the coordinate system $\Sigma_o$, as illustrated in Figure 1, which is close to the object's center of gravity.

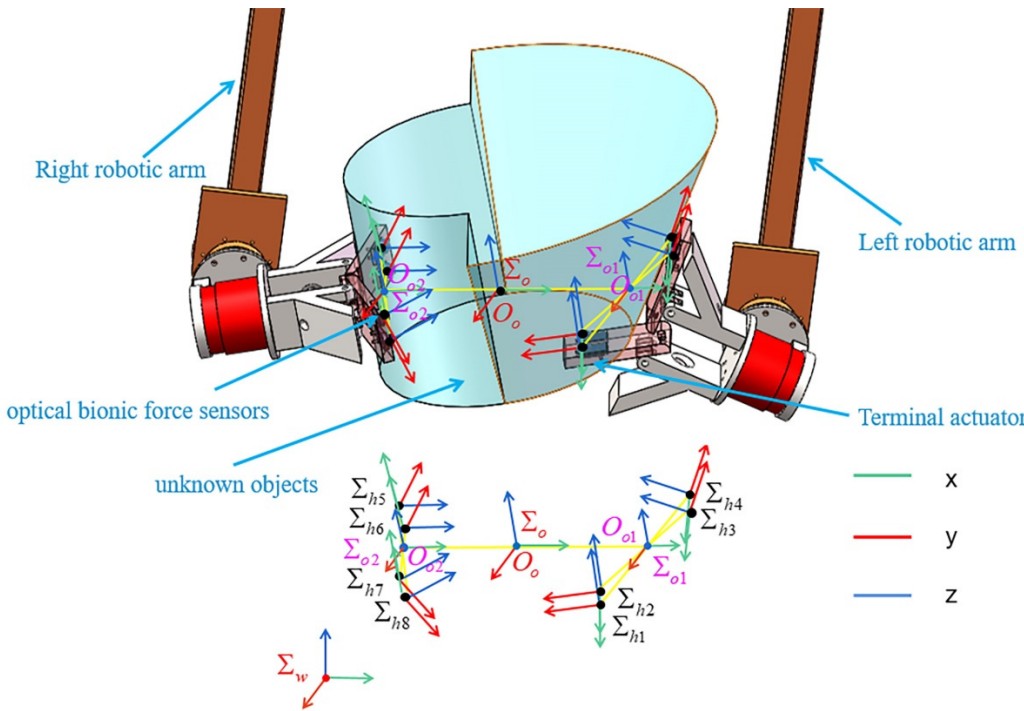

**Figure 1.** Four terminals equipped with two optical bionic force sensor contact points cooperate to clamp and stably handle heavy objects of any shape.

A virtual rod defines the force space control vector in a multi-terminal multi-contact-point collaborative system [23]. The contact point of this paper is a three-dimensional optical bionic force sensor (Figure 1). The coordinate system $\Sigma_{hi}(i = 1, \ldots, n)$ represents the contact point coordinate system ($n = 8$). Nevertheless, the proposed method can be applied to more contact points and terminals.

$\Sigma_{o1}$ and $\Sigma_{o2}$ represent the planar center coordinate system formed by the four coordinate system origins at the contact points of the two terminals installed on a single robotic arm.

In this paper, $O_o$ is set at the midpoint of the connecting line between $O_1$ and $O_2$. The virtual rod $^{w}l_{hi}$ from $O_{hi}$ to $O_o$ is equivalent to a rigid rod structure fixed at the contact point. The virtual rod top coordinate system $\Sigma_{hi}(i = 1, \ldots, 8)$ initially coincides with the object coordinate system $\Sigma_o$.

The force vector generated at the top of $^{w}l_{hi}$ is defined as

$$^{w}h_{li} \equiv \begin{bmatrix} ^{w}F_{li}^{\mathrm{T}} & ^{w}N_{li}^{\mathrm{T}} \end{bmatrix}^{T} \tag{1}$$

where $^{o}F_{li}$ and $^{o}N_{li}$ are the force and moment applied at the tip of the virtual rod $i$, respectively. The prefix $w$ indicates that the vector is defined in the robot's coordinate system $\Sigma_w$. The vector $^{w}h_{li}$ is calculated from the force and moment applied to the object by the contact terminal $i$. The sum of the combined forces on an object $^{w}F_o$ and moment $^{w}N_o$ is calculated from the sum of $^{w}h_{li}$:

$$^{w}h_o \equiv \begin{bmatrix} ^{w}F_{o}^{\mathrm{T}} & ^{w}N_{o}^{\mathrm{T}} \end{bmatrix}^{T} = G^{w}q_l \tag{2}$$

where $G \equiv [I_6 \ I_6 \ I_6 \ \ldots]$, $^{w}q_l \equiv \begin{bmatrix} ^{w}h_{l1}^{\mathrm{T}} & ^{w}h_{l2}^{\mathrm{T}} & ^{w}h_{l3}^{\mathrm{T}} & \cdots & ^{w}h_{ln}^{\mathrm{T}} \end{bmatrix}^{T}$, and $I_6$ is a 6 * 6 identity matrix with a rank of 6 because the matrix $G$ maps $n * 6$ dimensional vectors to six-dimensional

vectors. Therefore, its null space range is $(6 * n - 6)$-dimensional, with $n = 8$ in this paper. We define V as the null space basis of $G$. The general interpretation of Equation (2) is

$$^w q_l = G^- \, ^w h_o + V \, ^w h_{\mathrm{m}} \tag{3}$$

where $^w h_{\mathrm{m}}$ is an arbitrary $(6 * n - 6)$-dimensional vector corresponding to V and $G^-$ is the generalized inverse matrix of $G$.

Note that $V^w h_{\mathrm{m}}$ belongs to the null space of $G$. Thus, $^w h_m$ corresponds to the internal force/moment that can be controlled separately without affecting the net force. In addition, V and $^w h_m$ are not fixed, which is important for the follow-up force-tracking control.

Equation (3) can be rewritten as

$$^w q_l = [G^- \; V] \begin{bmatrix} ^w h_o \\ ^w h_{\mathrm{m}} \end{bmatrix} = U^w h$$

$$U \equiv [G^- \; V] \in \mathbb{R}^{6n * 6n}, {}^w h \equiv \begin{bmatrix} ^w h_o \\ ^w h_{\mathrm{m}} \end{bmatrix} \in \mathbb{R}^{6n} . \tag{4}$$

where $^w h$ is a generalized force vector. The internal forces $^w h_m$ can be expressed as a $(n-1)6$ dimensional force vector set:

$$^w h_{\mathrm{m}} \equiv \begin{bmatrix} ^w h_{\mathrm{r}1} \\ ^w h_{\mathrm{r}2} \\ \vdots \\ ^w h_{\mathrm{r}(n-1)} \end{bmatrix} \tag{5}$$

For a given $^w q_l$, the force/moment vector $^w h$ is obtained by solving Equation (4).

$$^w h = U^{-1} \, ^w q_l \tag{6}$$

Next, we introduce stable clamping and handling with multi-contact points in inner and outer working spaces, utilizing the controlling vectors and the required coordinate systems both simultaneously and independently. The physical meaning of this setup is summarized as follows. Matrix $G$ maps the forces exerted by the multiple terminals and contact points to the external forces of the object. Matrix V is the null space basis of $G$, and matrix U maps a set of external and internal forces to the forces of multi-terminal contact points.

### 2.2. Simplification and Transformation of the Internal Force Space

Next, we discuss the stable clamping internal force of the multi-point-contact collaborative handling system proposed in this paper. When more than two contact points conduct a collaborative clamp, it is difficult to determine the null space basis V proposed in Section 2.1 because V does not represent the intuitive meaning of the internal force/moment. Hence, we write Equation (6) as follows:

$$
\begin{aligned}
^w h_o &= {}^w h_{l1} + {}^w h_{l2} + \ldots + {}^w h_{l8} \, , \\
^w h_{\mathrm{r}1} &= c_{1,1}{}^w h_{l1} + c_{1,2}{}^w h_{l2} + \ldots + c_{1,n}{}^w h_{l8} \, , \\
^w h_{\mathrm{r}2} &= c_{2,1}{}^w h_{l1} + c_{2,2}{}^w h_{l2} + \ldots + c_{2,n}{}^w h_{l8} \, , \\
&\qquad\qquad\qquad \vdots \\
^w h_{\mathrm{r}7} &= c_{7,1}{}^w h_{l1} + c_{7,2}{}^w h_{l2} + \ldots + c_{7,8}{}^w h_{l8}.
\end{aligned}
\tag{7}
$$

Thus, Equation (6) becomes

$$^w h = \begin{bmatrix} ^w h_o \\ ^w h_{\mathrm{m}} \end{bmatrix} = \begin{bmatrix} G \\ C \end{bmatrix} {}^w q_l = U^{-1} \, ^w q_l \tag{8}$$

Once the seven internal force vectors ${}^w h_{r1} \ldots {}^w h_{r,7}$ are given, matrix $C$ is determined automatically. However, under the same net force requirement, the internal force vector ${}^w h_{r1} \ldots {}^w h_{r,7}$ has an infinite number of values. This paper determines the internal force direction of stable clamping according to the terminal model. Thus, the desired internal force for stable clamping is adaptively tracked under this set of internal force directions. Therefore, we extract the internal force vector Equation (9) from Equation (8):

$$ {}^w h_{\mathrm{m}} = C {}^w q_l \tag{9} $$

In fact, the workspace formula represented by Equations (1)–(9) can merge and decompose the force vectors at the contact points based on the task and model degrees of freedom. Therefore, in order to reduce the computational complexity and simplify the system, we can simplify the internal force workspace based on the degree of freedom of the terminal model and the characteristics of the task (this article does not involve tasks involving local or individual $\Sigma_{hi}(i = 1, \ldots, n)$ contact objects). According to the method we want to introduce in this paper, without loss of generality, we make the following reasonable assumptions:

1. No torque vector is formed at a single contact point.
2. No internal force is generated between two parallel contact points on a terminal.
3. No internal torque is generated between all contact points, so only internal forces are involved in the control based on reference trajectories.

Based on the above assumptions, the initial internal force workspace becomes a four-contact terminal force vector workspace. According to the system model depicted in Figure 2, the coordinate systems at the contact points $\Sigma_{h1}$ and $\Sigma_{h2}$ are merged into $\Sigma_{ha}$. The new coordinate direction remains unchanged, while the origin changes to $O_{ha}$ at the midpoint of the connected line between $O_{h1}$ and $O_{h2}$. Furthermore, as $\Sigma_{ha}$ is a new cooperation terminal, its force vector is determined by the force vector combination at the contact point of $\Sigma_{h1}$ and $\Sigma_{h2}$. We merge $\Sigma_{h3}$ and $\Sigma_{h4}$ to form $\Sigma_{hb}$; $\Sigma_{h5}$ and $\Sigma_{h6}$ to form $\Sigma_{hc}$; and $\Sigma_{h7}$ and $\Sigma_{h8}$ to form $\Sigma_{hd}$ in the same way. As presented in Figure 2, the force vector is simplified similarly. Therefore, the force vector applied to the object by the eight terminal contact points ${}^w q_h \equiv \left[ {}^w h_{h1}^T \; {}^w h_{h2}^T \; {}^w h_{h3}^T \ldots {}^w h_{h8}^T \right]^T$ described in Section 2.1 becomes ${}^w q_h \equiv \left[ {}^w h_{ha}^T \; {}^w h_{hb}^T \; {}^w h_{hc}^T \; {}^w h_{hd}^T \right]^T$, and accordingly, the force/moment vector at the tip of the virtual rod ${}^w q_l \equiv \left[ {}^w h_{l1}^T \; {}^w h_{l2}^T \; {}^w h_{l3}^T \ldots {}^w h_{l8}^T \right]^T$ becomes ${}^w q_l \equiv \left[ {}^w h_{la}^T \; {}^w h_{lb}^T \; {}^w h_{lc}^T \; {}^w h_{ld}^T \right]^T$. Thus, Equation (7) becomes

$$
\begin{aligned}
{}^w h_o &= {}^w h_{la} + {}^w h_{lb} + {}^w h_{lc} + {}^w h_{ld} \\
{}^w h_{r1} &= c_{1,1}{}^w h_{la} + c_{1,2}{}^w h_{lb} + c_{1,3}{}^w h_{lc} + c_{1,4}{}^w h_{ld} \\
{}^w h_{r2} &= c_{2,1}{}^w h_{la} + c_{2,2}{}^w h_{lb} + c_{2,3}{}^w h_{lc} + c_{2,4}{}^w h_{ld} \\
{}^w h_{r3} &= c_{3,1}{}^w h_{la} + c_{3,2}{}^w h_{lb} + c_{3,3}{}^w h_{lc} + c_{3,4}{}^w h_{ld}
\end{aligned}
\tag{10}
$$

Defining the direction of the intuitive force while handling an object with a human hand is trivial. Therefore, the following internal force group is used as an intuitive force given to the expected internal force group:

$$
\begin{aligned}
{}^w h_{r1} &= \tfrac{1}{2}\left( {}^w h_{la} - {}^w h_{lb} \right) \\
{}^w h_{r2} &= \tfrac{1}{2}\left( {}^w h_{lc} - {}^w h_{ld} \right) \\
{}^w h_{r3} &= \tfrac{1}{2}\left( {}^w h_{la} + {}^w h_{lb} \right) - \tfrac{1}{2}\left( {}^w h_{lc} + {}^w h_{ld} \right)
\end{aligned}
\tag{11}
$$

Thus, matrix $C$ in Equation (11) is determined as the intuitive expectation value matrix $C_d$ as follows:

$$
C_d = \begin{bmatrix} 0.5I_6 & -0.5I_6 & 0_6 & 0_6 \\ 0_6 & 0_6 & 0.5I_6 & -0.5I_6 \\ 0.5I_6 & 0.5I_6 & -0.5I_6 & -0.5I_6 \end{bmatrix} \tag{12}
$$

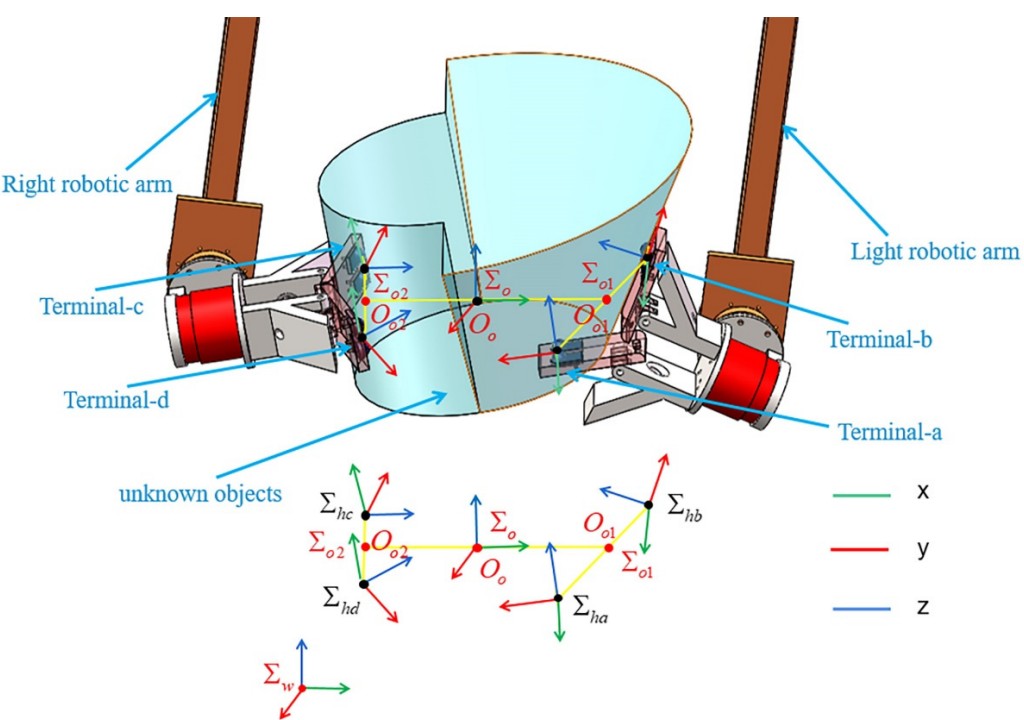

**Figure 2.** Simplified system workspace.

The simplified generalized internal force vector $^wq_{md} \equiv \begin{bmatrix} ^wh_{r1}^T {}^wh_{r2}^T {}^wh_{r3}^T \end{bmatrix}^\mathrm{T}$ is the expected generalized internal force vector. Thus, Equation (11) becomes

$$^wq_{md} \equiv C_d * {}^wq_l \tag{13}$$

Although the expected internal force vector $^wq_{md}$ determined in Equation (13) is intuitive and in line with conventional thinking from the perspective of understanding, it significantly increases the difficulty of setting the parameters. It even makes it impossible to achieve favorable control from a control perspective. Therefore, we perform a favorable transformation of the internal force workspace appropriate for the control method proposed in this paper. Considering the robustness of the multi-terminal adaptive scheme introduced in this paper, it is necessary to determine one of the terminals and control its absolute motion to enable the other terminals to control the force tracking of the relative motion of this terminal.

Note that the internal force vector does not contribute to the object's motion but represents the mechanical stress exerted on the object. There are infinite representations of internal force vectors as null space vectors. The time-varying complex internal force space is difficult to determine, and it is difficult to accurately control the collaborative task of manipulating unknown objects using multiple terminals. One of this paper's contributions is modelling the multiple terminals as an accurate controllable space.

Based on the model in Equation (10), the selected internal force group is used as the control variables and converted to meet the control needs. In this paper, the desired internal force group in Equation (13) and the internal force group from the slave terminal position to the master terminal are converted in the following equation to achieve the control goal. The internal force group from the slave terminal to the master terminal is defined as follows:

$$\begin{aligned}
^wh_{rb} &= \tfrac{1}{2}(^wh_{la} - {}^wh_{lb}), \\
^wh_{rc} &= \tfrac{1}{2}(^wh_{la} - {}^wh_{lc}), \\
^wh_{rd} &= \tfrac{1}{2}(^wh_{la} + {}^wh_{ld}).
\end{aligned} \tag{14}$$

Matrix $C$ determined by Equation (14) will be named $C_e$, the internal command force parameter.

$$C_e = \begin{bmatrix} 0.5I_6 & -0.5I_6 & 0_6 & 0_6 \\ 0.5I_6 & 0_6 & -0.5I_6 & 0_6 \\ 0.5I_6 & 0_6 & 0_6 & -0.5I_6 \end{bmatrix} \qquad (15)$$

When $^w q_{me} \equiv \begin{bmatrix} ^w h_{rb}^T {}^w h_{rc}^T {}^w h_{rd}^T \end{bmatrix}^{\mathrm{T}}$ is determined as a generalized command internal force vector, then Equation (14) becomes:

$$^w q_{me} \equiv C_e * {}^w q_l \qquad (16)$$

If we determine the expected internal force set, the command internal force group is uniquely determined since we can perform the following conversion between Equations (13) and (16).

$$\begin{aligned} C_d^- \, {}^w q_{md} &= {}^w q_l \\ {}^w q_{me} &= C_e C_d^- \, {}^w q_{md} \end{aligned} \qquad (17)$$

where $C_d^-$ is the generalized inverse of $C_d$. The transformation in Equation (17) is critical for introducing force-tracking adaptive control.

The relationship between the expected and the command internal force in Equation (17) is represented in Figure 3, where the force applied by each terminal on the object is transferred to the origin of the object coordinate system through a virtual rod. $^w F_{r1}, {}^w F_{r2}, {}^w F_{r3}$ represents the intuitive expected internal force extracted from Equation (11) in $^w h_{r1}, {}^w h_{r2}, {}^w h_{r3}$, and $^w F_{rb}, {}^w F_{rc}, {}^w F_{rd}$ is the internal instruction force—the key control quantity in this article. Due to the four terminals in zero space, the combined external force $^w F_o$ applied to the object is controlled near a fixed value according to the internal force in changing the demand of the object's movement. It is easy for operators to regularly observe the internal force $^w F_{r1}, {}^w F_{r2}, {}^w F_{r3}$. Through observation, we conclude that it meets the demand for stable clamping of objects. However, as a control variable, achieving internal force-tracking tasks is difficult because a tracked end is a tracking end itself, which poses significant difficulties in the controller design. Due to the four terminals in zero space, the combined external force F applied to the object is controlled near a fixed value according to the internal force changing the demand of the object's movement. In order to solve such difficulties, this article proposes the instruction of the internal force $^w F_{rb}, {}^w F_{rc}, {}^w F_{rd}$ (extracted from $^w h_{r1}, {}^w h_{r2}, {}^w h_{r3}$ in Equation (14)) through Equations (11)–(17). As shown in Figure 3, $^w F_{rb}, {}^w F_{rc}, {}^w F_{rd}$ is the internal force between one terminal and the other terminals, which is crucial for force-tracking control. If this end is determined as the absolute-motion control object, it will solve the problem of the tracked end itself being a tracking end. This greatly reduces the uncertainty of control. As shown in Equation (17), $^w F_{rb}, {}^w F_{rc}, {}^w F_{rd}$ is uniquely determined when $^w F_{r1}, {}^w F_{r2}, {}^w F_{r3}$ is determined. In Section 3, we will track a fixed $^w F_{rb}, {}^w F_{rc}, {}^w F_{rd}$ on a one-dimensional holding line while determining $^w F_{r1}, {}^w F_{r2}, {}^w F_{r3}$.

### 2.3. Absolute Relative Motion Control

Human hands can hold and handle unknown large objects of any shape in any way stably and flexibly. It is worth highlighting how complex time-varying internal forces and motion are regulated through perceptual intelligence. In this paper, we simulate the ability of humans to develop a multi-terminal robot control algorithm. In the above human process, a person perceives that any part of their palm is in contact with the object and can be consciously determined as the master terminal, while other parts of their palm in contact with the object can be determined as the slave terminal. Moreover, the slave terminal clamp (increase the internal force) to the master terminal is determined by consciousness to stabilize the clamping. Furthermore, it is important to keep the relative movement between the master terminal and the slave terminal within a small stable range (time-varying force and motion tracking). Similarly, the task space of the robot can be broadly divided into the

combined motion of the absolute motion of the master terminal and the relative motion of the slave terminal.

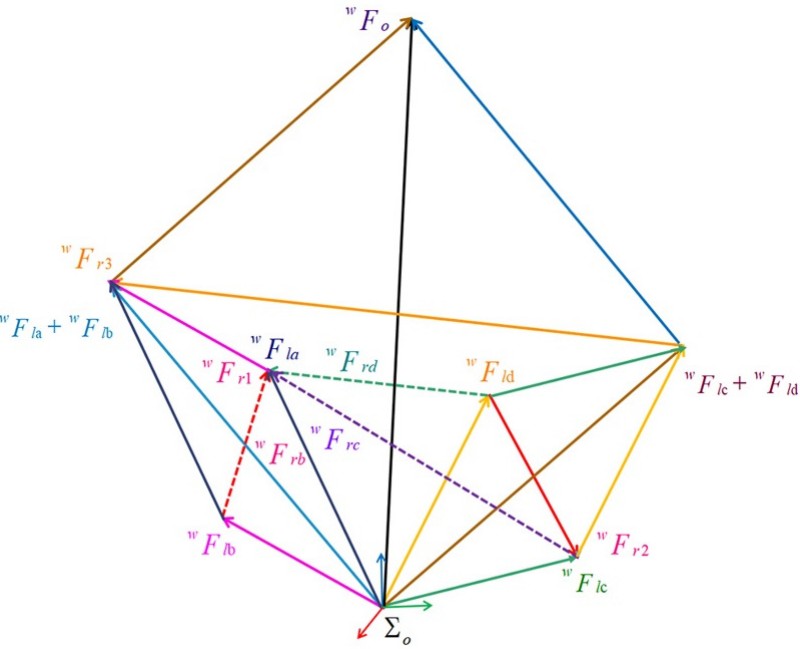

**Figure 3.** Relationship between the expected internal force and the commanded internal force.

The absolute motion is performed by the master terminal and is designed to achieve accurate and robust tracking of the target reference trajectory. On the other hand, relative motion is conducted by the slave terminal, designed to accept the movement of the master terminal. Thus, the so-called closed kinematic chain constraints can be satisfied. The overall framework is shown in Figure 4; in this paper, $\Sigma_{ha}$ presented in Figure 2 is identified as the master terminal, Terminal-a, responsible for absolute motion control. The other $\Sigma_{hj}$, $(j = b, c, d)$ are slave terminals, named Terminal-j $(j = b, c, d)$, and are responsible for relative motion control.

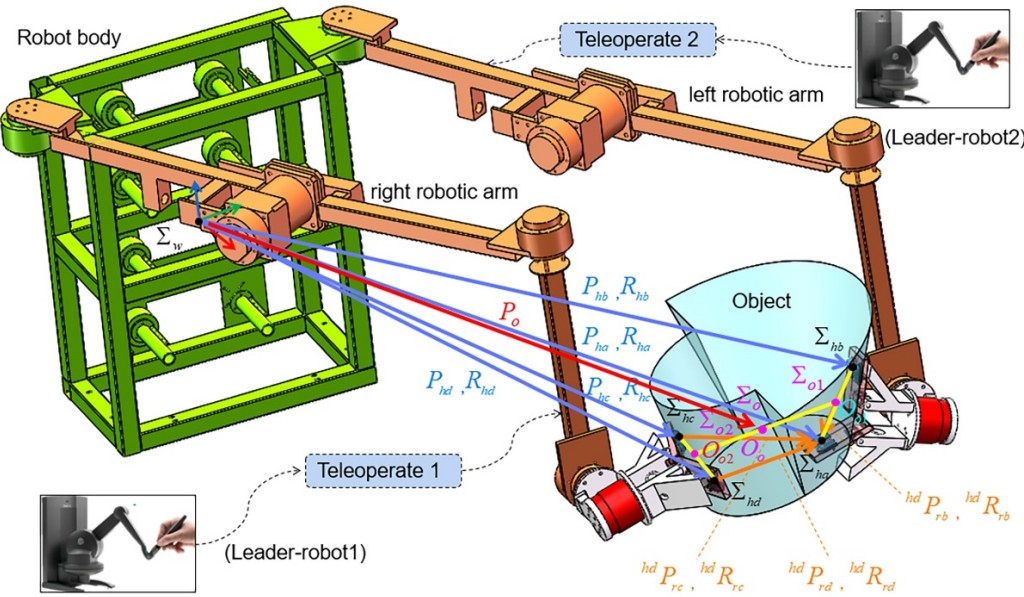

**Figure 4.** The overall framework.

Figure 2 illustrates the relationship between each terminal and object motion in the robot coordinate system, which can be explained with the following equation:

$$
\begin{cases}
{}^{w}P_{hi} = {}^{w}P_o - {}^{w}R_{hi}\,{}^{hi}l_i \\
{}^{w}\dot{p}_{hi} = {}^{w}\dot{p}_o - \left({}^{w}R_{hi}\,{}^{hi}l_i\right)^{\times}{}^{w}\omega_o \\
{}^{w}\omega_{hi} = {}^{w}\omega_o
\end{cases}
\tag{18}
$$

where ${}^{w}p_{hi}(i = a, b, c, d)$ is the position vector of the simplified four terminals in the robot coordinate system, $(\cdot)^{\times}$ is the cross product executed by the oblique symmetry matrix operator, ${}^{w}R_{hi}(i = a, b, c, d)$ represents the transformation matrix of each terminal coordinate system relative to the robot coordinate system, ${}^{w}\dot{p}_o$ and ${}^{w}\omega_o$ are the translation velocity and angular velocity of the object in the robot coordinate system, respectively, and ${}^{hi}l_i$ is the virtual rod represented by the terminal coordinate system.

The relative position ${}^{hj}p_{rj}$ and orientation ${}^{hj}Q_{rj}$ of the three slave terminals, Terminal-j ($j = b, c, d$), and the one master terminal, Terminal-a, in Equation (18) are expressed in the slave terminal coordinate system as follows:

$$
{}^{hj}p_{rj} = {}^{w}R_{hj}{}^{T}\left({}^{w}p_{ha} - {}^{w}p_{hj}\right)
\tag{19}
$$

$$
{}^{hj}Q_{rj} = {}^{w}R_{hj}^{T}\,{}^{w}Q_{ha}
\tag{20}
$$

In an ideal state, there is no relative motion between each terminal, so the relative position ${}^{hj}p_{rj}$ and orientation ${}^{hj}Q_{rj}$ should remain constant.

The robot coordinate system is the default and will be omitted in subsequent equations. At time t, the expected translation velocity and angular velocity corresponding to the object are expressed by $\dot{p}_o^{d}(t)$ and $\omega_o^{d}(t)$, respectively. According to Equation (18), we can obtain the expected translation velocity $\dot{p}_{ha}^{d}(t)$ and angular velocity $\omega_{ha}^{d}(t)$ of the master terminal as follows:

$$
\dot{p}_{ha}^{d}(t) = \dot{p}_o^{d}(t) + \left(R_{ha}(t)\,{}^{hi}l_a\right)^{\times}\omega_o^{d}(t)
\tag{21}
$$

$$
\omega_{ha}^{d}(t) = \omega_o^{d}(t)
\tag{22}
$$

$$
e_{pha}(t) = p_{ha}^{d}(t - T) - p_{ha}(t)
\tag{23}
$$

$$
e_{Qha}(t) = Q_{ha}^{d}(t - T) - Q_{ha}(t)
\tag{24}
$$

where $e_{pha}(t)$ represents the absolute position error of the master terminal, $e_{Qha}(t)$ indicates the absolute direction error of the master terminal, $T$ is the sampling time step, and $p_{ha}^{d}(t - T)$ is the command position of the last sampling time. $p_{ha}(t)$ is the measurement position of the current time, and $Q_{ha}^{d}(t - T)$ is the desired direction at the last sampling time, with $Q_{ha}(t)$ representing the direction measured at the current moment. Note that only $Q_{ha}^{d}(t - T)$ and $Q_{ha}(t)$ are consistent, while $e_{Qha}$ will be zero.

Once the Cartesian velocity of the master terminal is established, the desired velocity of the slave actuator can be determined by

$$
\dot{p}_{hj}^{d}(t) = v_{ha}(t) - \left({}^{hj}p_{ha}^{d}\right)^{\times}\omega_{ha}(t)
\tag{25}
$$

$$
\omega_{hj}^{d}(t) = \omega_{ha}(t)
\tag{26}
$$

where $j = b, c, d$, $v_{ha}(t)$ is the master terminal translation velocity and ${}^{hj}p_{ha}^{d} = {}^{hj}p_{ha}(t_0)$ is the relative position initialized from the terminal coordinate system extracted from the terminal gesture by (19).

Consider relative errors:

$$e_{prj}(t) = {}^{w}R_{hj}(t)\left[{}^{hj}p_{ha}^{d} - {}^{hj}p_{ha}(t)\right] \tag{27}$$

$$e_{Qrj}(t) = {}^{w}R_{hj}(t)\left[{}^{hj}Q_{ha}^{d} - {}^{hj}Q_{ha}^{d}(t)\right] \tag{28}$$

where $e_{prj}(t)$ indicates the relative position error of each slave actuator under the robot coordinate system and $e_{Qrj}(t)$ indicates the relative direction error of each slave terminal under the robot coordinate system. Similarly, ${}^{hj}p_{ha}^{d} = {}^{hj}p_{ha}(t_0)$ and ${}^{hi}Q_{ha}^{d} = {}^{hi}Q_{hi}(t_0)$ represent the expected relative position and direction expressed in the slave actuator coordinate system, respectively.

With the above closed-chain constraint method, we can quantitatively calculate the absolute and relative motion of the four terminals accordingly for the desired position and orientation of the object. The advantage of using master–slave closed-chain constraint control is that multi-terminal systems can perform collaborative tasks without knowing the information about the shape, size, and stiffness of the object.

## 3. Multi-Terminal Adaptive Motion

Each terminal in the system detailed in this paper interacts with the object in point contact, so only the translational force is considered without considering the rotational torque in the force of a single terminal on the object. Multi-terminal collaborative forces provide the rotational torque of an object.

The intrinsic transformation relationship between multiple terminals for the desired internal force direction and the command internal force system was established in Section 2.2 (Equation (17)). Furthermore, controlling the absolute relative motion with closed-chain constraints is described in Section 2.3 (Equations (18)–(20)). Thus, we create an adaptive reference trajectory control model for time-varying force-tracking multi-slave terminals in a specified internal force direction.

### 3.1. Problem Description

Since the exact position and motion of the object during the time-varying motion are uncertain, and the stiffness $k$ of the object is unknown, completing time-varying internal force tracking directly from the terminal relative motion control is impossible. Therefore, in this paper, we propose an adaptive scheme for tracking the expected internal forces of multiple terminals.

The internal command force is defined in Section 2.2. In the following derivation, the slave Terminal-j ($j = b, c, d$) will track the relative position ${}^{hj}P_{rj}$ between its coordinate system and the master terminal Terminal-a, i.e., the reference position trajectory of the direction $\overrightarrow{O_{hj}O_{ha}}$ to achieve the desired command internal force. The time-varying force tracking for this specified direction can be transformed into a virtual spring two-stage motion, as depicted in Figure 5 $p_{rje}$, ($j = b, c, d$) is defined as the measurement position of the slave terminal in the $\overrightarrow{O_{hj}O_{ha}}$ direction, and $p_{rjo}$ is defined as the position of the initial contact point between the slave terminal and the object and the initial $\overrightarrow{O_{hj}O_{ha}}$, $p_{rjo} = p^{hje}$, ($j = b, c, d$). Extracting the expected command internal force $F_{rj}^{d}$, ($j = b, c, d$) from the generalized command internal force vector ${}^{w}q_{me}$ in Equation (17), the internal force error can be expressed as $\Delta F_{rj} = F_{rj} - F_{rj}^{d}$. In position control mode, assume $p_{rje}^{d} = p_{rje}$, and the stiffness of an unknown object is $k_{e}$. Then, the internal command force can be expressed by

$$F_{rj} = k_{e}\left(p_{rjo} - p_{rje}\right) \tag{29}$$

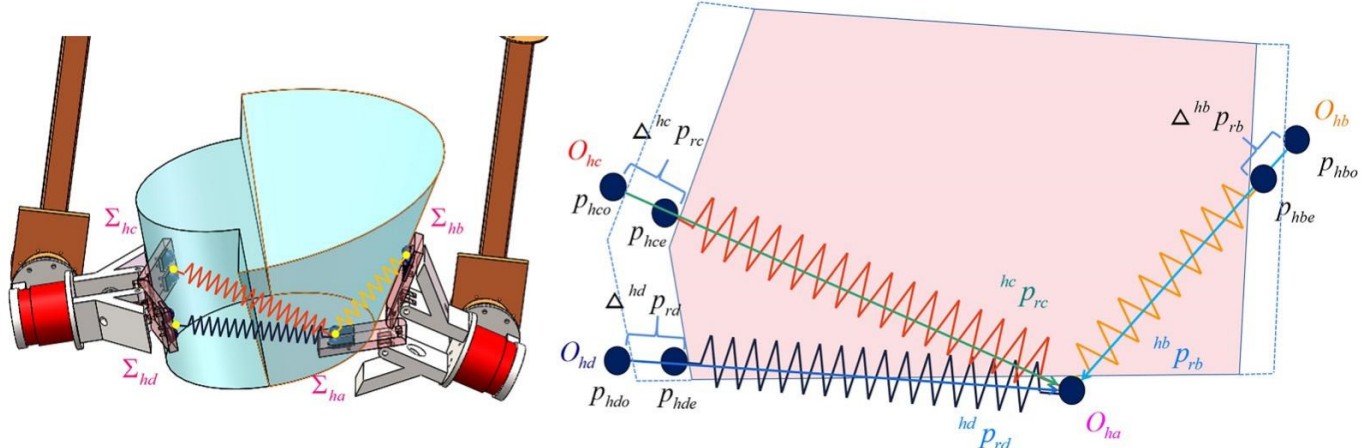

**Figure 5.** Time-varying force tracking converted into virtual spring two-stage motion.

Equation (29) shows that if it is known that the reference position $p_{rj}^r, (j = b, c, d)$ of the direction $\overrightarrow{O_{hj}O_{ha}}$ depends on the exact position of the object $p_{rjo}, (j = b, c, d)$ and the exact stiffness of the object $k$, then it can be calculated from the required and desired internal force $p_{rj}^r, (j = b, c, d)$.

$$p_{rj}^r = p_{rjo} - \frac{F_{rj}^d}{k_e} \tag{30}$$

However, the object's exact position during the handling process is uncertain in actual control. In Figure 5 $p_{rjo}, (j = b, c, d)$ is the initial contact point position, and in the subsequent control process, it completely depends on the dynamic environment which is composed of the master terminal and the objects to calculate. The time-varying internal force is kept near the fixed value in this dynamic environment and thus fully considers the impact of the master terminal and objects' motions on the internal force. As a result of the stiffness of unknown objects $k$, the error of the model parameters, the control accuracy, and the existence of uncertain factors (such as noise), the trajectory $p_{rjo}, (j = b, c, d)$ is uncertain, so it is impossible to directly calculate the relative motion reference position trajectory of the slave terminal.

Note: The internal force workspace transformation in Section 2.2 proposed the direction of the internal command force, which meets the control requirements. In the same sense, each slave terminal coordinate system origin $O_{hj}(j = b, c, d)$ tracks the desired position at the holding line $\overrightarrow{O_{hj}O_{ha}}$ pointing to the origin of the master terminal coordinate system $O_{ha}$. Therefore, next, we will focus on the one-dimensional variables at the holding line $\overrightarrow{O_{hj}O_{ha}}$ of $p_{hjo}, (j = b, c, d)$ and $p_{hje}, (j = b, c, d)$ in dynamic and uncertain environments.

### 3.2. Unknown Motion Estimation of the Internal Command Force Direction

Next, we will use the internal command force $F_{rj}, (j = b, c, d)$ to estimate the relative position vector $^{hj}P_{rj}$ shown in Figure 5, i.e., the unknown time-varying position $p_{rjo}$ of the direction $\overrightarrow{O_{hj}O_{ha}}$, which is critical for $p_{rj}^r$ presented in Section 3.3. In general, we model $p_{rjo}$ with the following time-varying trajectories:

$$\begin{aligned}
p_{rb0}(t) &= b_0 + b_1 t + b_2 t^2 + \ldots + b_{n-1} t^{n-1} \\
p_{rc0}(t) &= c_0 + c_1 t + c_2 t^2 + \ldots + c_{n-1} t^{n-1} \\
p_{rd0}(t) &= d_0 + d_1 t + d_2 t^2 + \ldots + d_{n-1} t^{n-1}
\end{aligned} \tag{31}$$

where $t$ is time $b_{n-1}$, $c_{n-1}$, $d_{n-1}$, where $n = 1, 2, 3 \cdots$, which is considered a constant for a certain period. For the conciseness of the subsequent formulas, we will set

$g_{n-1} = b_{n-1}$, $c_{n-1}$, $d_{n-1}$. Thus, in Equation (31), the corresponding velocity and acceleration will be:

$$p_{rjo}(t) = g_0 + 2g_1 t + \ldots + g_{n-1} t^{n-1},$$
$$\dot{p}_{rjo}(t) = g_1 + 2g_2 t + \ldots + (n-1)g_{n-1} t^{n-2}, \ (g = b, c, d), (j = b, c, d) \qquad (32)$$
$$\ddot{p}_{rjo}(t) = 2g_2 + \ldots + (n-1)(n-2)g_{n-1} t^{n-3}.$$

Based on the model in Equation (32), the estimate of $p_{rjo}$ is described as follows:

$$\hat{p}_{rjo} = \hat{g}_0 + \hat{g}_1 t + \hat{g}_2 t^2$$
$$\dot{\hat{p}}_{rjo} = \hat{g}_1 + 2\hat{g}_2 t \qquad (33)$$
$$\ddot{\hat{p}}_{rjo} = 2\hat{g}_2$$

where $\hat{\ast}$ is an estimated value of $\ast$, defined according to Equation (33).

$$\hat{F}_{rj} = \hat{k}(\hat{p}_{rjo} - p_{rje}) \qquad (34)$$

where $\hat{F}_{rj}$ and $\hat{k}$ are the estimated command internal forces and the Hooke coefficient of the object, respectively. By subtracting Equations (34) and (29), we obtain:

$$\hat{F}_{rj} - F_{rj} = \hat{k}\left(\hat{g}_0 + \hat{g}_1 t + \hat{g}_2 t^2 - p_{rje}\right) - k\left(g_0 + g_1 t + g_2 t^2 - p_{rje}\right)$$
$$= \begin{bmatrix} p_{rje} & 1 & t & t^2 \end{bmatrix} H_j. \ (g = b, c, d), \ (j = b, c, d) \qquad (35)$$

In this equation,

$$H_j = \begin{bmatrix} -\hat{k} + k \\ \hat{k}\hat{g}_0 - kg_0 \\ \hat{k}\hat{g}_1 - kg_1 \\ \hat{k}\hat{g}_2 - kg_2 \end{bmatrix} \qquad (36)$$

Equation (36) represents the estimated error for the unknown total kinetic parameters set. Then, we design the adaptive renewal law of $H_j$ based on Lyapunov theory:

$$\dot{H}_j = \begin{bmatrix} -\dot{\hat{k}} \\ \dot{\hat{k}}\,\hat{g}_o + \hat{k}\,\dot{\hat{g}}_o \\ \dot{\hat{k}}\,\hat{g}_1 + \hat{k}\,\dot{\hat{g}}_1 \\ \dot{\hat{k}}\,\hat{g}_2 + \hat{k}\,\dot{\hat{g}}_2 \end{bmatrix} = -\Gamma_j^{-1} \begin{bmatrix} p_{rje} \\ 1 \\ t \\ t^2 \end{bmatrix} (\hat{F}_{rj} - F_{rj}), \ (j = b, c, d), \ (g = b, c, d) \qquad (37)$$

Thus, the adaptive renewal law of $\hat{k}$ and $\hat{g}_i$ in Equation (37) can be determined as

$$\dot{\hat{k}} = \gamma_{j1}^{-1} p_{rje}(\hat{F}_{rj} - F_{rj}), \ j = b, c, d. \qquad (38)$$

$$\dot{\hat{g}}_{i-1} = -\frac{1}{\hat{k}}\left(\gamma_{j,i+1}^{-1} t^{i-1}(\hat{F}_{rj} - F_{rj}) + \dot{\hat{k}}_e \hat{g}_{i-1}\right), \ g = b, c, d. \ i = 1, 2, 3. j = b, c, d. \qquad (39)$$

where the renewal rate in the equation is $\gamma_{j,i}$. In Equation (37), $\Gamma_j = \mathrm{diag}\left[\gamma_{j,1}, \ \gamma_{j,2}, \ \gamma_{j,3}, \ \gamma_{j,4}\right]$, $j = b, c, d$ is a positive definite matrix. In the following equation, we prove the adaptive

renewal law. From Equations (36)–(39), we conclude that the force estimation error of $\hat{F}_{rj} - F_{rj}$ converges to 0.

Next, we construct a Lyapunov function:

$$V_j = \frac{1}{2} H_j^T \Gamma_j H_j \, , j = b, c, d \tag{40}$$

Taking the derivative of Equation (40) concerning time will yield

$$\dot{V}_j = H_j^T \Gamma_j \dot{H}_j \tag{41}$$

Taking the transposition on both sides of the Equation (35) will provide

$$\left(\overset{\wedge}{F}_{rj} - F_{rj}\right)^T = H_j^T \begin{bmatrix} p_{rje} \\ 1 \\ t \\ t^2 \end{bmatrix} . \, j = b, c, d. \tag{42}$$

Substituting the adaptive renewal law (37) into Equation (41) and considering Equation (42) provides

$$\dot{V}_j = -H_j^T \begin{bmatrix} p_{hj} \\ 1 \\ t \\ t^2 \end{bmatrix} (\hat{F}_{rj} - F_{rj}) = -(\hat{F}_{rj} - F_{rj})^T (\hat{F}_{rj} - F_{rj}). \tag{43}$$

Equation (43) shows that when there is a force estimation error of $\tilde{F}_{rj} = \hat{F}_{rj} - F_{rj}$, then $\dot{V}_j \leq 0$ and $V_j$ will decrease. Thus, when the renewal law in Equation (37) eventually leads to $t \to \infty$, then $\tilde{F}_{rj} \to 0$.

*3.3. Tracking the Generation of Reference Trajectories of Expected Command Internal Forces*

Next, we develop an algorithm to generate $p_{rj}^r$, $(j = b, c, d)$ and the reference trajectories $p_{rje}, j = b, c, d$ by estimating the uncertain motion of $P_{rjo}, j = b, c, d$ and the object's deformation $\triangle p_{rj}, j = b, c, d$, as depicted in Figure 4, so that the internal forces required to stabilize clamping are fixed around a certain value when the object is moved.

As shown in Figure 5, according to Equation (29), we set a required command internal force group:

$$F_{rj}^d = k \left( p_{rjo} - p_{rje}^d \right), j = b, c, d \tag{44}$$

This corresponds to the desired trajectory

$$p_{rje}^d = p_{rjo} - \frac{1}{k_e} F_{rj}^d, j = b, c, d \tag{45}$$

where $F_{rj}^d$ represents the desired command internal force and $p_{rje}^d$ represents the desired trajectory at the holding line $\overrightarrow{O_{hj}O_{ha}}$ from the terminal actuator Terminal-j, $(j = b, c, d)$. According to Section 3.2, if we estimate $p_{rjo}$ and $k$, we can obtain the expected trajectory of $p_{rje}^d$ using Equation (45), which will be used to compare with the upcoming impedance-model-based method.

In Equation (46), we define the target impedance model at the holding line $\overrightarrow{O_{hj}O_{ha}}$ from the terminal actuator Terminal-j $(j = b, c, d)$,

$$M\ddot{p}_{rje} + B\dot{p}_{rje} + K(p_{rje} - p_{rje}^d) = e_{rj} \, , j = b, c, d \tag{46}$$

where $M = \text{diag}[m_x, \, m_y, \, m_z]$, $B = \text{diag}[b_x, \, b_y, \, b_z]$, and $K = \text{diag}[k_x, \, k_y, \, k_z]$ are the mass, damping, and stiffness parameters of each slave terminal, respectively, in the robot coordi-

nate system $x, y, z$ directions. $e_{rj} = F_{rj} - F_{rj}^d$ is the internal command force-tracking error, and $F_{rj}^d$ is the expected command internal force.

According to Equation (29), we obtain:

$$\begin{aligned} p_{rje} &= -\tfrac{1}{k}F_{rj} + p_{rjo} \\ &= -\tfrac{1}{k}(F_{rj}^d + e_{rj}) + p_{rjo}, j = b, c, d \end{aligned} \tag{47}$$

Substituting Equation (47) into the impedance model (46) provides

$$\begin{aligned} M\ddot{e}_{rj} + B\dot{e}_{rj} &+ (K + kI_3)e_{rj} = \\ &-(MF_{rj}^d + B\dot{F}_{rj}^d + KF_{rj}^d) + k(M\ddot{p}_{rjo} + B\dot{p}_{rjo} + Kp_{rjo}) - kKp_{rje}^d \end{aligned} \tag{48}$$

where $I_3$ represents a $3 \times 3$ identity matrix. Since $F_{rj}^d$ is constant, Equation (48) becomes

$$\begin{aligned} M\ddot{e}_{rj} + B\dot{e}_{rj} &+ (K + kI_3)e_{rj} = \\ &-KF_{rj}^d + k(M\ddot{p}_{rjo} + B\dot{p}_{rjo} + Kp_{rjo}) - kKp_{rje}^d \, . \end{aligned} \tag{49}$$

For the force-tracking error dynamics in Equation (49), the steady-state error is

$$e_{rj,ss} = k(K + I_3 k)^{-1}(-\tfrac{1}{k}KF_{rj}^d + (M\ddot{p}_{rjo} + B\dot{p}_{rjo} + Kp_{rjo}) - Kp_{rje}^d) \tag{50}$$

Therefore, to eliminate the steady-state error $p_{rje}^d$, the desired trajectory of $p_{rje}$, $j = b, c, d$ must meet

$$p_{rje}^d = K^{-1}(M\ddot{p}_{rjo} + B\dot{p}_{rjo} + Kp_{rjo} - \tfrac{1}{k}KF_{rj}^d) \tag{51}$$

By replacing $p_{rjo}$ and $k$ in Equation (51) with the renewal law $\hat{p}_{rjo}$ and $\hat{k}$ established in Section 3.2, the reference trajectory of $p_{rje}$, $j = b, c, d$ will be

$$p_{rje}^r = \frac{1}{K}(M\ddot{\hat{p}}_{rjo} + B\dot{\hat{p}}_{rjo} + K\hat{p}_{rjo} - \frac{K}{\hat{k}}F_{rj}^d), j = b, c, d \tag{52}$$

As demonstrated in [24], we finally obtain the command internal force system $F_{rj} \rightarrow F_{rj}^d$, $(j = b, c, d)$ using $p_{rje}^r$, a reference trajectory of $p_{rje}, j = b, c, d$, to eliminate the steady-state error $e_{rj,ss}$.

In summary, we have obtained $\hat{F}_{rj} \rightarrow F_{rj} \rightarrow F_{rj}^d$, $(j = b, c, d)$, $\hat{F}_{rj} \rightarrow F_{rj}$ as described in Section 3.2 and $F_{rj} \rightarrow F_{rj}^d$ as described in Section 3.3. Hence, adaptively adjusting the internal forces based on multi-holding lines by referencing trajectories $p_{rje}^r$, $(j = b, c, d)$ based on motion estimation and impedance models causes the object's clamping command internal force to converge to the desired value. As transformed and defined in Section 2.2, the user provides the intuitive meaning and expected clamping internal force group, and then the system transforms it into a command internal force group. In this way, force-tracking control from the slave to the master is realized.

## 4. Experiments

### 4.1. Experimental Description

The experiments were carried out using two self-made six-degrees-of-freedom manipulators, with two actuator terminals at the ends and robotic arms with two parallel contact points at the actuator terminals. The specific dimensions of the robotic arms are shown in Table 1. A three-dimensional bionic optical force sensor was installed at each contact point to measure the contact force. All sensors and motors were connected to a PC via

a hub. The absolute and relative motion were controlled with classic proportional gain, and an adaptive time-varying force-tracking control algorithm was used to integrate the overall controller based on the force/bit hybrid control model. Due to its length, mature algorithms are not mentioned. For further details, the reader is referred to [19].

**Table 1.** The relevant dimensions of the robotic arm.

| Module | Length (mm) | Width (mm) |
| --- | --- | --- |
| Link 1 | 480 | |
| Link 2 | 480 | |
| Link 3 | 373.5 | |
| Link 4 | 126.5 | |
| Link 5 | 110.5 | |
| Link 6 | 54.5 | |
| Terminal | 100 | 50 |
| Sensor center distance | 25 | |
| Terminal axis spacing | 57 | |

To prove the performance of the proposed algorithm, a multi-terminal system was designed to manipulate two objects with different shapes and stiffnesses:

1. The first manipulated object is softly wrapped, and its surface is soft and uneven.
2. The second manipulated object is a container with a hard and flat surface.

The geometry of the manipulated object and its related dimensions, position, and stiffness are arbitrary and unknown. In addition, experiments were performed under unknown nonideal conditions (e.g., deformable objects of any shape; arbitrary clamping; and arbitrary position, direction trajectories, sliding, and friction), which are suitable for nonstructural applications such as rescue. Under the above experimental setup, the reliability and robustness of the proposed multi-terminal adaptive collaboration method are verified.

The software architecture is based on a robot operating system (ROS), the communication method is RS services, and the implementation of the control module benefits from a set of software packages that implement the control algorithms. When autonomous control is realized, it can communicate with the robot through the read–write port. The key posture of the entire operation task of the software package is shown in Figure 6. Figure 7a,b present the expected experimental trajectories. The soft-body package exhibits a change in position and direction, while the container has only a position translation, where the translational trajectory is the same as for the soft-body package.

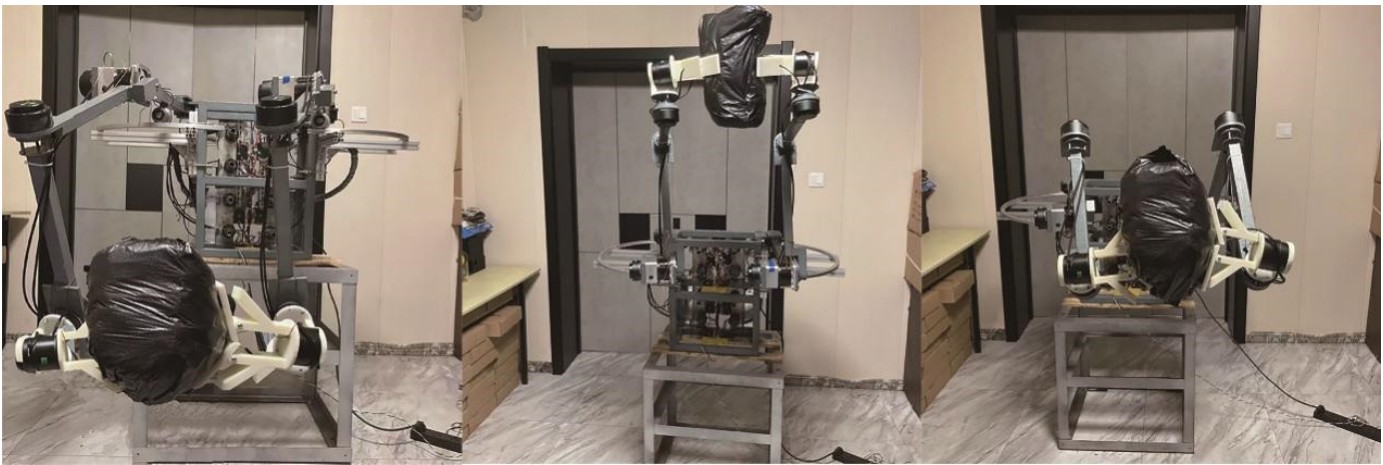

**Figure 6.** The key pose of the entire operation task for the soft-package.

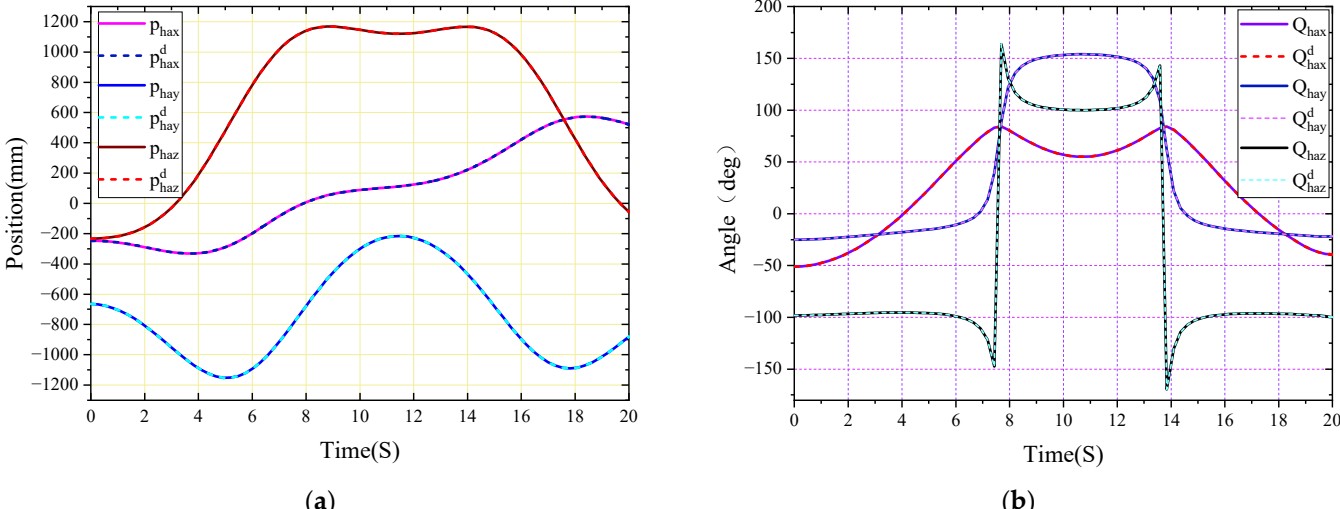

**Figure 7.** Absolute motion control: (**a**) absolute position trajectory; and (**b**) absolute direction trajectory.

The reference trajectory Terminal-j, $(j = b, c, d)$ requires some time to stabilize and smooth the reference trajectory by $p_{rje}^{r1} = p_{rje}^{r}(1 - e^{-\mu t^{\eta}}), j = b, c, d$. Therefore, starting from 0, $p_{rje}^{r1}$ is gradually dominated by $p_{rje}^{r}$. In all cases, we set $\mu = 0.02, \eta = 2$.

Regarding the remaining parameters, the initial values are $\hat{k} = 10, \dot{\hat{k}} = 0,$
$\left[\hat{b}_0, \hat{c}_0, \hat{d}_0\right] = [0.55, 0.55, 0.55], \left[\dot{\hat{b}}_0, \dot{\hat{c}}_0, \dot{\hat{d}}_0\right] = [0,0,0], \left[\hat{b}_1, \hat{c}_1, \hat{d}_1\right] = [0,0,0],$
$\left[\dot{\hat{b}}_1, \dot{\hat{c}}_1, \dot{\hat{d}}_1\right] = [0,0,0], \left[\hat{b}_2, \hat{c}_2, \hat{d}_2\right] = [0,0,0],$ and $\left[\dot{\hat{b}}_2, \dot{\hat{c}}_2, \dot{\hat{d}}_2\right] = [0,0,0].$ MBK is given as M= diag[0.01, 0.01, 0.01], B= diag[0.1, 0.1, 0.1], and K= diag[5, 5, 5].

### 4.2. Soft Package Manipulation Experiment

#### 4.2.1. Motion Control Analysis

The soft package was an elastomer with an uneven and irregular surface which could be manipulated to any shape. The mechanical response to the applied force was nonlinear. The soft-body wrapping could quickly absorb and regenerate the force during the translation and rotation of multi-terminal gripping objects.

The key poses for the entire experimental process are presented in Figure 6. In the first stage, the operator visually judged the shape of the object through remote operation commands and decided the initial clamping posture of the four terminals, ensuring that each contact point (3D biomimetic optical force sensor) had a contact force of about 1N with the object. Based on the first stage, the second stage triggered the proposed command internal force adaptive tracking algorithm. Each slave terminal actuator from the end effector was close to the master terminal along the direction of its hold line $\overrightarrow{O_{hj}O_{ha}}$, $(j = b, c, d)$, until it adaptively obtained the desired command internal force. In the third stage, the object's specified position trajectory and direction trajectory were manipulated under the joint operation of absolute, relative, and reference trajectory control of unknown adaptive motion.

The absolute motion position trajectory is depicted in Figure 7a, and the direction trajectory is illustrated in Figure 7b, revealing that the main end actuator trajectory was smoother, suggesting that the master terminal actuator tracked the desired trajectory well. The absolute position and direction errors are shown in Figure 8a,b. Due to the existence of

unknown uncertainties, such as coordinate calibration error, numerical error and Kinematics uncertainty, the absolute error has a small and acceptable oscillation near zero.

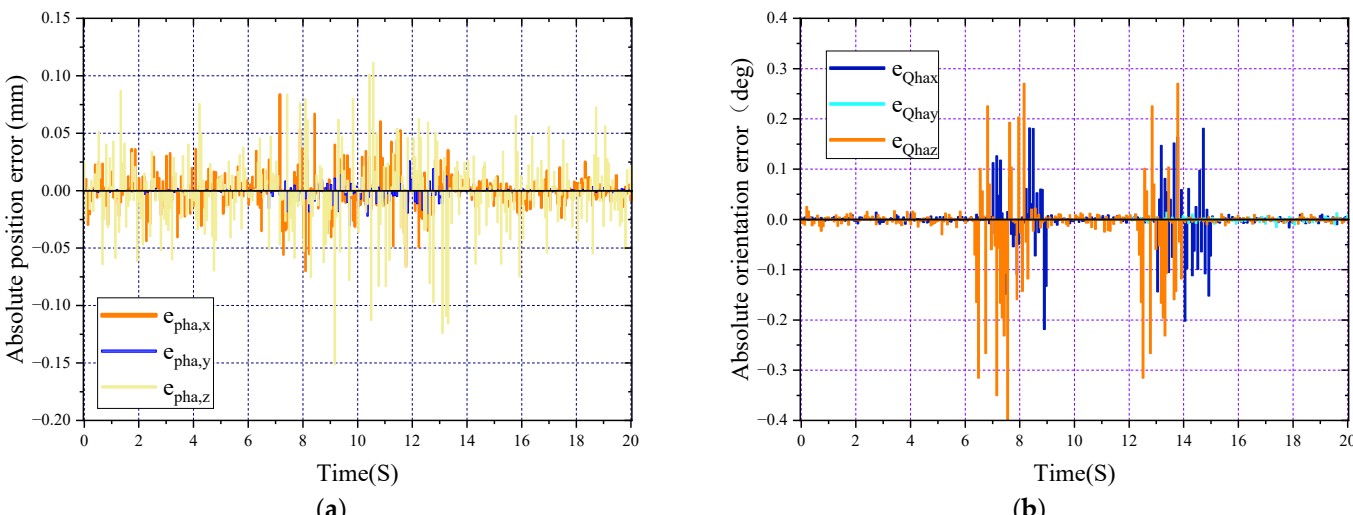

**Figure 8.** Absolute motion error: (**a**) absolute position trajectory error; and (**b**) absolute direction trajectory error.

The projection of each slave relative position trajectory on the xz and yz planes of the robot's coordinate system is illustrated in Figure 9. The slave trajectory highlights that after triggering the adaptive tracking algorithm of the internal command force in the second stage, each reference trajectory was generated by adaptively estimating unknown parameters from the terminal; each slave terminal was close to the master terminal along the holding line, tracking the trajectory of the internal command force. At first, the holding line $\overrightarrow{O_{hj}O_{ha}}$, $(j = b, c, d)$, which is the vector $^{hb}p_{rb}, ^{hc}p_{rc}, ^{hd}p_{rd}$ in Figure 5, varied and oscillated in a certain range of directions within its coordinate system. Thus, special attention should be paid to the fact that the internal forces traced on the holding line do not necessarily satisfy the desired internal forces, assuming an artificially given initial relative position between each terminal, including frictional requirements.

Therefore, the expected value of the commanded internal force $^{w}F_{rb}, ^{w}F_{rc}, ^{w}F_{rd}$ directly given without considering the expected internal force $^{w}F_{r1}, ^{w}F_{r2}, ^{w}F_{r3}$ in Figure 3 was insufficient to meet the system stability requirements. If a reasonable expected internal force $^{w}F_{r1}, ^{w}F_{r2}, ^{w}F_{r3}$ is given according to Equation (17), it is converted to a reasonable expected value of $^{w}F_{rb}, ^{w}F_{rc}, ^{w}F_{rd}$, avoiding the above instability. $\hat{k}$ and the unknown parameters introduced in Section 3 were estimated in a stable range around the holding line under reasonable desired commanded internal force conditions, with each slave end being under the control constraint of a constant relative position and the feedback force controller. Thus, the relative trajectory was well maintained with the help of the reference trajectory. The relative position and direction trajectory errors are depicted in Figure 10a,b. Under reasonable initial position and expected command internal force conditions, each slave adaptively estimated the unknown parameters of the absolute position, and the direction errors are presented in Figure 8a,b. Due to unknown uncertainties, such as coordinate calibration error, numerical error, and kinematic uncertainty, the absolute error has a small and acceptable oscillation around zero.

A reference trajectory was generated and gradually stabilized to the final relative position under the influence of the reference trajectory. The subsequent process of unknown motion estimation and tracking revealed that the relative position and direction error remained near zero, the position oscillation amplitude was less than 1 mm, and the directional oscillation amplitude was below one degree. These errors may combine many factors, such as self-made model errors, unknown uncertain dynamic environment estimations, and renewal rates. When the two slave terminals far from the main terminal changed rapidly toward the

absolute motion direction due to the object's inertia, there was a large return deviation and other positional and directional errors. The maximum deviation reached 5 mm, which is believed to be related to parameter adjustments. Therefore, future work will focus on solving such problems by optimizing the parameters, model, and algorithm. Overall, the effectiveness of the multi-terminal collaboration algorithm proposed in this paper is proven.

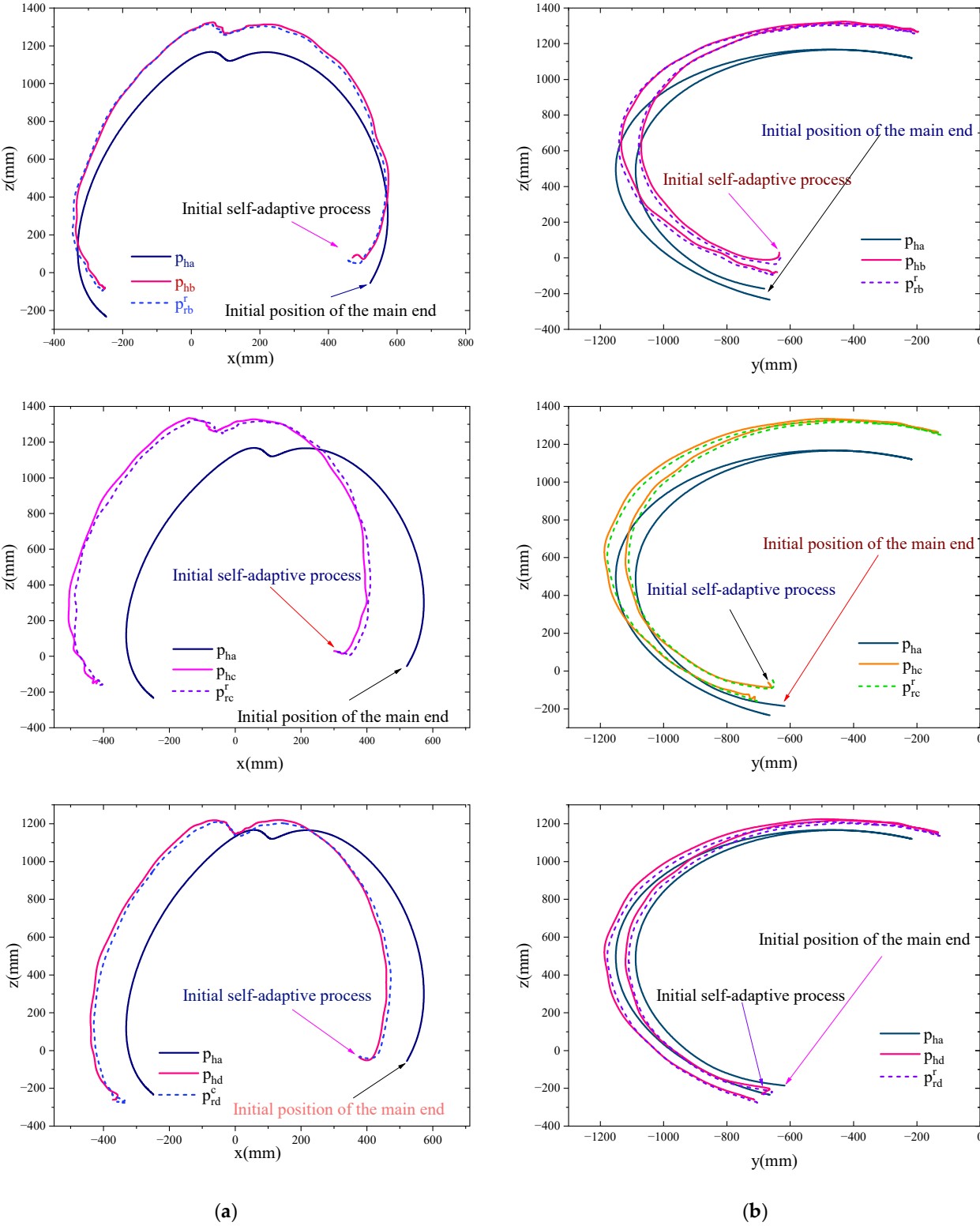

(**a**)          (**b**)

**Figure 9.** (**a**) Relative position trajectory in the xz plane projection of each slave. (**b**) Relative position trajectory in the yz plane projection of each slave.

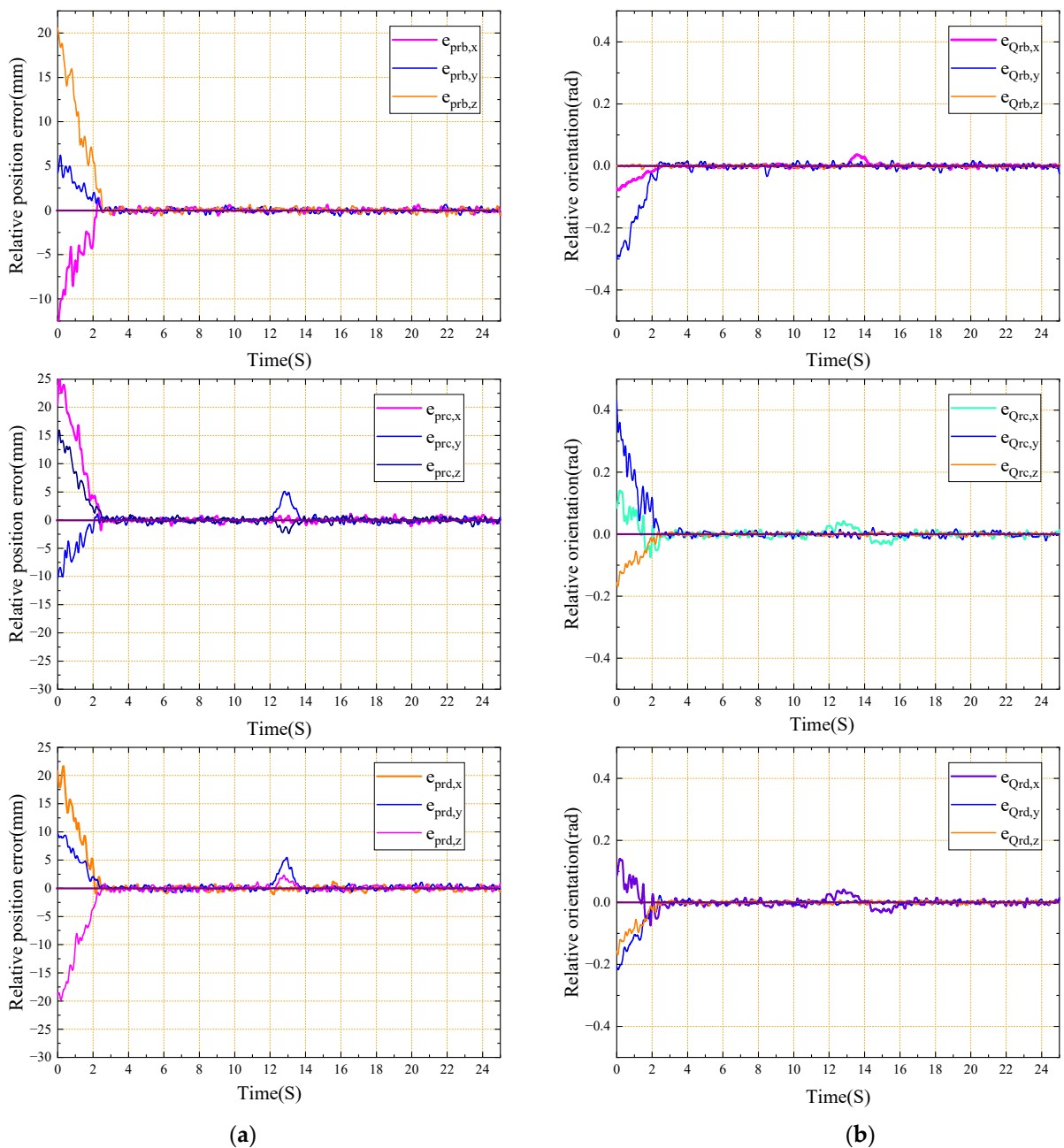

**Figure 10.** Relative motion error: (**a**) relative position trajectory error of each slave; and (**b**) relative direction trajectory error of each slave.

### 4.2.2. Internal Force Tracking Analysis

The following experiments aimed to prove the effectiveness and superiority of the proposed method for arbitrary trajectory handling of large objects with unknown information in an arbitrary way using multi-terminal actuators. These experiments focused on simplification, transformation, and adaptive tracking methods for complex internal force spaces.

Here, the operator observed the desired internal force $[^wF_{r1}, ^wF_{r2}, ^wF_{r3}] = [-7N, -5N, -24N]$, which satisfied the clamping motion requirement in a given intuitive sense and converted it into the desired commanded internal force $[^wF_{rb}, ^wF_{rc}, ^wF_{rd}] = [-7N, -13N, -18N]$ employing Equation (17), as presented in Section 4.2.1. The above operation is essential for multi-terminal arbitrary clamping and handling of unknown objects of arbitrary shape due to the reasonableness of the desired

internal force $[{}^{w}F_{r1}, {}^{w}F_{r2}, {}^{w}F_{r3}] = [-7N, -5N, -24N]$ enhancing the possibility of tracking the desired command internal force on the holding line $\overrightarrow{O_{hj}O_{ha}}$, $(j = b, c, d)$. Failing the task because of slippage or insufficient internal force between the slave ends was also avoided. Figure 3 fully illustrates this point. In addition, the internal forces exerted on the object from the three slave terminals are shown in Figure 11a. When the reference position began to be adaptively updated in the direction of the hold line, the initial overshoot of each slave terminal was about 25%. The internal command force stabilized at the desired command internal force group $[-7N, -13N, -18N]$ in about 7 s. During the subsequent operation, the internal force was always controlled near the expected command internal force with the help of the reference trajectory. Note that the initial trajectory time was calculated from the beginning of the second stage of the entire experiment, and the internal force history was recorded from the first stage. The direction of ${}^{w}F_{rb}, {}^{w}F_{rc}, {}^{w}F_{rd}$ was along the holding line $\overrightarrow{O_{hj}O_{ha}}$, $(j = b, c, d)$, and Figure 3 highlights that the force ${}^{w}F_{la}$ at the master end was time-varying. In this case, adaptively tracking the direction of ${}^{w}F_{rb}, {}^{w}F_{rc}, {}^{w}F_{rd}$ to the holding line $\overrightarrow{O_{hj}O_{ha}}$, $(j = b, c, d)$ and the desired value under the constraint that the relative motion direction of the multiple slave to the master end remains constant is a unique contribution of this paper.

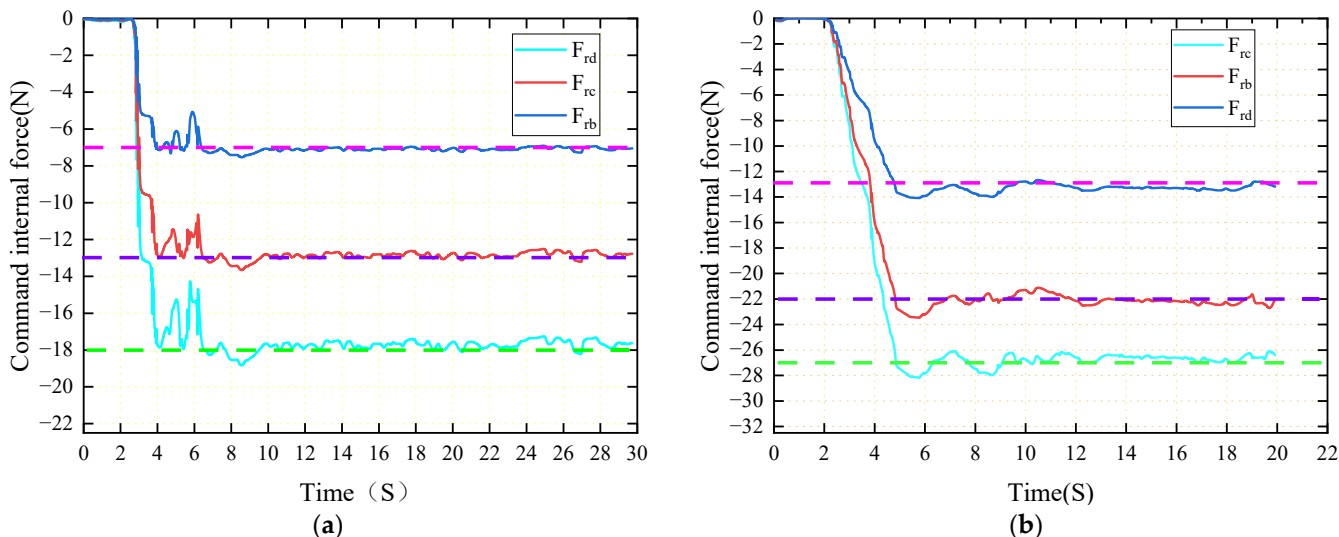

**Figure 11.** Internal force-tracking error: (**a**) history of the internal force group of the soft manipulation package; and (**b**) history of the internal force group of the manipulation container.

Figure 11a illustrates the error response. Despite the uncertainty of the object stiffness and position, the tracking command internal force error of each slave was less than 1 N (6%). When adaptive control was disabled, the system's overshoot increased. One of the main goals of these controllers is to avoid force overshoot during the contact phase while maintaining internal force-tracking errors. In the following experiments, this phenomenon will become even more critical. Meanwhile, in this paper, we also compare the effectiveness and applicability of our proposed method with other types of control algorithms in force tracking of uncertain and unknown moving targets, especially in multi-terminal environments. With support from references [18,19], as well as numerous similar studies, we constructed a comparative table (Table 2). From the table, it can be observed that the adaptive impedance control method based on reference trajectories established in this paper is more competitive in terms of the error response, parameter settings requirements, and computational complexity of the multi-terminal model.

**Table 2.** Comparison of force-tracking performance in dynamic and uncertain environments.

| Methods | Steady-State Force Error | Overshoot | Setting Time | Requirements for Stability Parameters | Applicability to Multi-Terminal Collaboration |
|---|---|---|---|---|---|
| Adaptive impedance control based on a reference trajectory | <1N | 15% | 5 s | Low | Strong |
| Pure motion control | × | 76% | 36 s | Non-convergence | × |
| Constant impedance control | <8 N | 43% | 16 s | Non-convergence | × |
| Variable impedance control | <3 N | 32% | 13 s | High | Weak |

### 4.3. Manipulating Containers

In practice, the object stiffness varies significantly depending on the materials. To prove that the proposed method is suitable for anthropomorphic clamping and handling of objects of any stiffness and shape, a container weighing up to 3 kilograms containing debris was manipulated. Unlike the soft package, the container's surface was hard, flat, and conical in shape. The high stiffness of the container meant that adaptive force tracking became more difficult.

To demonstrate the environmental adaptability and user-friendliness of the proposed method, all control parameters were the same as in the previous experiments. Since the container was loaded with debris and the container's surface was smooth, the model contact point was not enough to meet the friction required by rotation. Note that only the translational operation of the previous experiments' position trajectory was executed in this experiment. If there are enough contact points, the arbitrary rotation task can be completed. The key poses of this experiment are depicted in Figure 12.

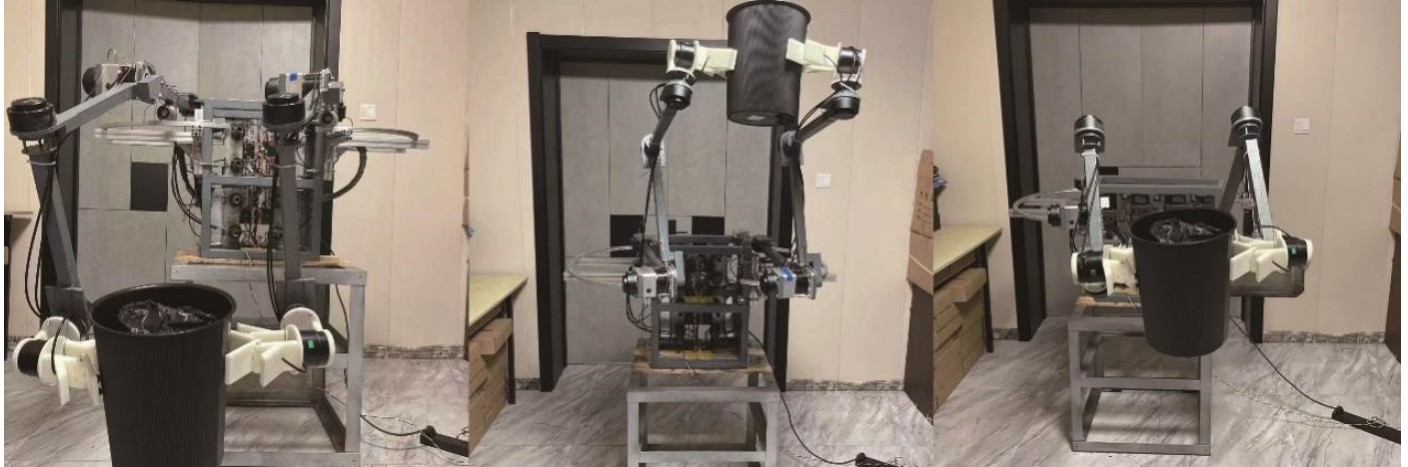

**Figure 12.** Key poses of translational high-stiffness container.

The corresponding expected internal force was $[^{w}F_{r1}, {}^{w}F_{r2}, {}^{w}F_{r3}] = [-13N, -5N, -36N]$, and the expected internal command force group was $[^{w}F_{rb}, {}^{w}F_{rc}, {}^{w}F_{rd}] = [-13N, -22N, -27N]$, with historical records illustrated in Figure 11b. These records infer that the adaptive process was longer than the specified internal force-tracking error and the time amplitude was larger. The tracking command internal force error of each slave was less than 2 N (8%).

The absolute position trajectory is depicted in Figure 7a. Similarly, the relative position trajectory and position tracking error are presented in Figures 13 and 14, respectively. Note that due to the translational manipulation, there was no significant deviation from the relative position after the internal force stabilized, as shown in Figure 10b. However, due to the high stiffness and smooth surface, the position fluctuated greatly and tracking the expected command internal force in the initial stage required more time.

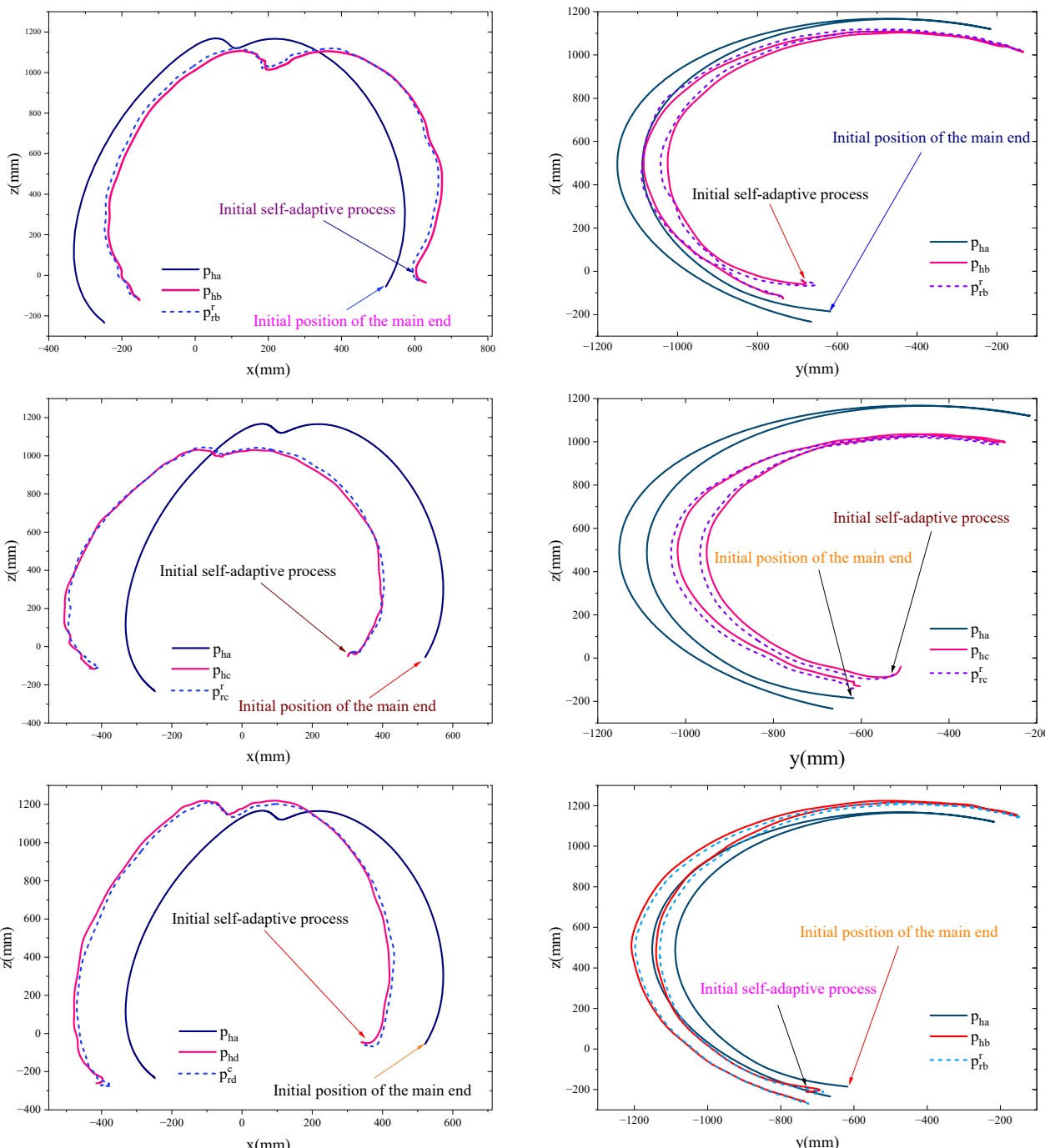

**Figure 13.** The xz plane and yz plane projection of the container translational relative motion trajectory.

So far, the effectiveness and accuracy of the proposed adaptive reference trajectory method have been proven. This method allows for manipulating unknown objects with arbitrary shapes through multi-terminal collaboration, which tracks the expected command internal force and relative motion control algorithm in the specified internal force direction. Compared with pure motion control and constant impedance control, ineffectiveness in manipulating unknown objects has been demonstrated in [13], and thus no comparative experiments will be conducted in this paper. Thus, we conclude that:

1. The proposed method effectively extracts and regulates holding lines for complex internal force spaces during multi-terminal force closure collaborations and maintains contact forces.

2. The motion estimation in the proposed multi-terminal collaboration method is important for dealing with multi-terminal time-varying motion.

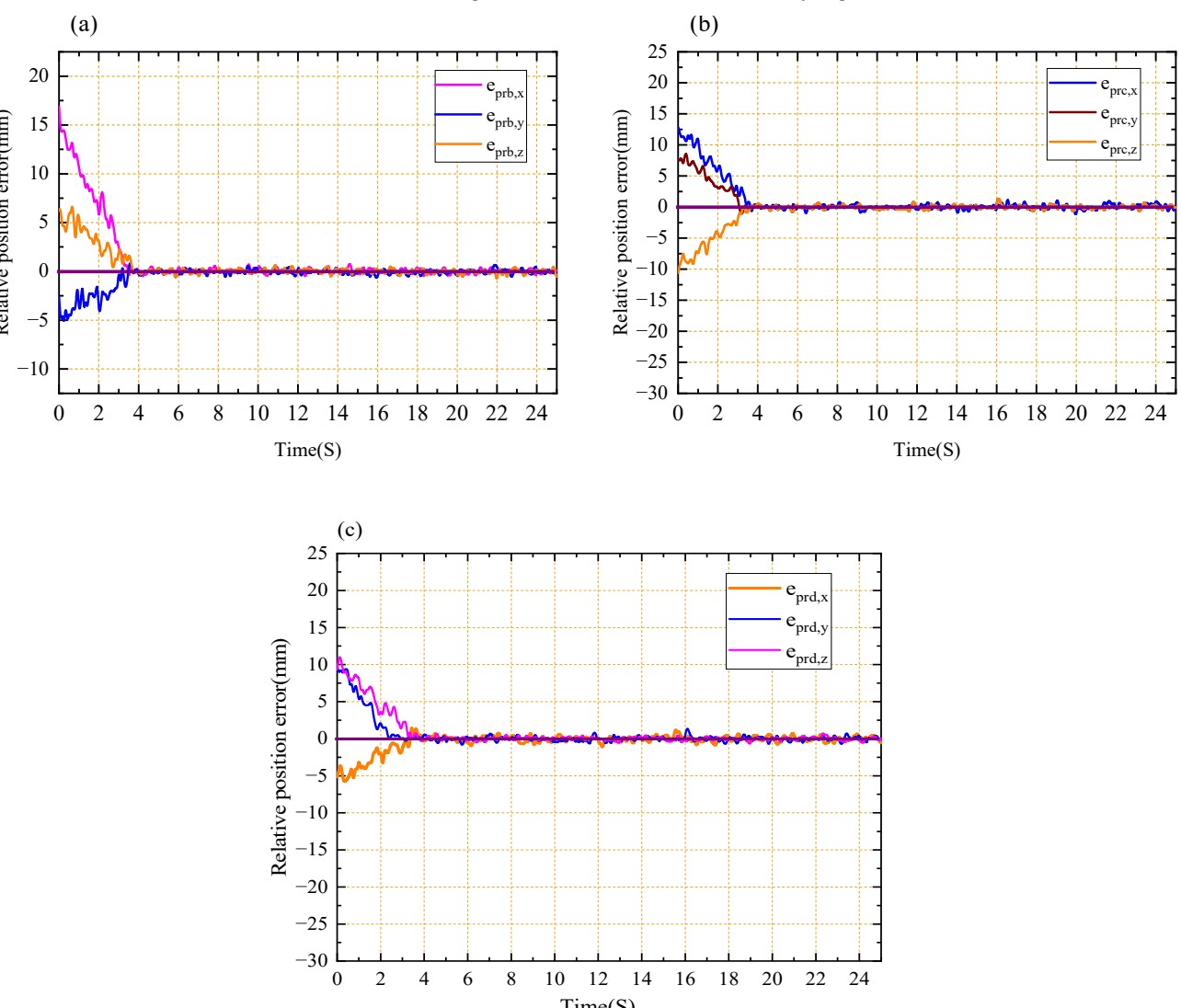

**Figure 14.** (**a**) Position trajectory error of b slave terminal in container translation.(**b**) Position trajectory error of c slave terminal in container translation. (**c**) Position trajectory error of d slave terminal in container translation.

## 5. Conclusions

In this paper, a multi-terminal actuator adaptive cooperation method is proposed to manipulate unknown objects for a system with closed-chain motion and dynamic uncertainty of the multi-arm, multi-terminal actuator and multi-contact sensing model. Drawing on the intelligent behavior of human clamping and handling of unknown objects, simplification of and conversion to effective control is established for the complex time-varying internal force working space during multi-terminal collaborative clamping and handling of unknown objects. Based on Lyapunov theory, an algorithm for multiples slaves to estimate the object stiffness and motion uncertainty in the direction of the internal command force is proposed. Moreover, impedance control is used to generate a reference trajectory for multi-slave ends, maintaining the desired command internal force on a reasonable holding line to track the motion of the master terminal. In this way, the stability and accuracy of the entire control system are realized. The proposed method is simple, stable, and robust to an object's shape, position, and stiffness changes.

Experiments on a two-arm robot system with multi-terminal, multi-contact sensing points verified that the proposed control method shows good position and force-tracking performance when the multi-terminal actuator clamps manipulate unknown objects of any shape or stiffness in any way. The specific results, such as experimental data, show that the relative position oscillation amplitude of each slave position $p_{hb}$, $p_{hc}$, $p_{hd}$ under the reference trajectory of motion estimation after tracking the expected command internal force to a stable position is less than 1 mm (except for in the case of a violent change in the direction of motion of the main end) and the directional oscillation amplitude is less than 1 degree. For objects with a low stiffness, the internal force error of each command from the terminal is less than 1 N (6%). Additionally, the internal force error of the tracking command for objects with a high stiffness is less than 2 N (8%). In addition, the method proposed in this article focuses on the collaborative force closure collaboration of multiple terminals (rather than the internal force closure collaboration of fixtures) and is limited to the model being formulated with eight sensing contact points and four terminals. However, the proposed algorithm can be extended to the collaborative force closure collaboration of $n$ multiple sensing contact points (tactile research) and $n$ multiple terminal actuators. Future work will focus on developing anthropomorphic robots with multiple terminals, utilizing self-adjusting expected internal forces based on sliding perception and more functional and nuanced models, and manipulating objects with complex and variable external forces. Accordingly, the new algorithms for these purposes are expected to have more requirements and lead to more difficulties.

**Author Contributions:** Conceptualization, Z.W.; methodology, Z.W. and J.D.; software, Z.W.; validation, F.S.; writing—original draft preparation, Z.W. and F.S.; writing—review and editing, J.D.; supervision, J.D.; funding acquisition, J.D. All authors have read and agreed to the published version of the manuscript.

**Funding:** This research was funded by the National Natural Science Foundation of China, grant number 62275066.

**Institutional Review Board Statement:** Not applicable.

**Informed Consent Statement:** Not applicable.

**Data Availability Statement:** Not applicable.

**Acknowledgments:** The authors would like to gratefully acknowledge the National Natural Science Foundation of China and would like to thank the editors and reviewers for their valuable comments and constructive suggestions. The authors would like to express their gratitude to Edit Springs (https://www.editsprings.cn, accessed on 5 May 2023) for the expert linguistic services provided.

**Conflicts of Interest:** The authors declare no conflict of interest.

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
