# Peer review of "Model-Based Adaptive Collaboration of Multi-Terminal Internal Force Tracking"

_applsci, doi:10.3390/app13158672_

Round 1

Reviewer 1 Report

some typo errors are there in the paper need to be rectified.

Reviewer 2 Report

1. In line 142 change the literal “e”. This literal is used in different parts of the text and causes confusion

2. Plot the “O0, O1 and O2 on the free body diagram.

3. From line 226 to line 229. What method does it ensure, what you mention.

4. In section 2.3. Place a block diagram of how the entire process is constituted.

5. In equation 46 clearly define the matrix M, B and K.

6. In section 4.1. Place in a table the dimensions and values of both arms and the grippers (fingers).

7. On line 564. Place the communication that is used when the ROS software is used.

Correct some grammatical mistakes.

Round 2

Reviewer 2 Report

Much better, thank you

No comment